# Nanoemulsions of Clove Oil Stabilized with Chitosan Oleate—Antioxidant and Wound-Healing Activity

**DOI:** 10.3390/antiox12020273

**Published:** 2023-01-26

**Authors:** Sara Perteghella, Alice Garzoni, Alessandro Invernizzi, Milena Sorrenti, Cinzia Boselli, Antonia Icaro Cornaglia, Daniele Dondi, Simone Lazzaroni, Giorgio Marrubini, Carla Caramella, Laura Catenacci, Maria Cristina Bonferoni

**Affiliations:** 1Department of Drug Sciences, University of Pavia, Viale Taramelli 12, 27100 Pavia, Italy; 2Department of Public Health, Experimental and Forensic Medicine, Histology and Embryology Unit, University of Pavia, Via Forlanini 2, 27100 Pavia, Italy; 3Department of Chemistry, University of Pavia, Viale Taramelli 14, 27100 Pavia, Italy

**Keywords:** clove oil, nanoemulsions, chitosan oleate, antioxidant activity, wound healing

## Abstract

Clove oil (CO) is a powerful antioxidant essential oil (EO) with anti-inflammatory, anesthetic, and anti-infective properties. It can be therefore considered a good candidate for wound-healing applications, especially for chronic or diabetic wounds or burns, where the balance of reactive oxygen species (ROS) production and detoxification is altered. However, EOs require suitable formulations to be efficiently administered in moist wound environments. Chitosan hydrophobically modified by an ionic interaction with oleic acid (chitosan oleate, CSO) was used in the present work to stabilize CO nanoemulsions (NEs). The dimensions of the NE were maintained at around 300 nm as the volume distribution for up to six months, and the CO content did not decrease to under 80% over 4 months, confirming the good stabilizing properties of CSO. The antioxidant properties of the CO NE were evaluated in vitro by a 2,2-diphenil-2-picrylhydrazyl hydrate (DPPH) assay, and in fibroblast cell lines by electron paramagnetic resonance (EPR) using *α*-phenyl-*N*-tert-butyl nitrone (PBN) as a spin trap; a protective effect was obtained comparable to that obtained with *α*-tocopherol treatment. In a murine burn model, the ability of CO formulations to favor macroscopic wound closure was evidenced, and a histological analysis revealed a positive effect of the CO NE on the reparation of the lesion after 18 days. Samples of wounds at 7 days were subjected to a histological analysis and parallel dosage of lipid peroxidation by means of a thiobarbituric acid-reactive substances (TBARS) assay, confirming the antioxidant and anti-inflammatory activity of the CO NE.

## 1. Introduction

Clove oil (CO) is an essential oil (EO) obtained from *Syzygium aromaticum* L., Myrtaceae, containing mainly eugenol (about 70%) and a lower percentage (about 10%) of *β*-caryophyllene [1], which is endowed with a variety of biological properties, including antioxidant and anti-inflammatory effects [2,3,4].

From the perspective of topical applications for tissue repair, both these activities are of special interest for supporting the healing process. In healing, four phases are generally recognized: hemostasis, inflammation, proliferation, and remodeling. After injuries, the inflammation phase has a positive role because it contributes to the removal of necrotic tissue and pathogens through leukocyte chemotaxis, while the production of reactive oxygen species (ROS) and nitric oxide that occurs during this phase by neutrophils facilitates pathogen inactivation and removal. However, an excess of the inflammatory response in terms of intensity and duration and the presence of ROS characterizes non-healing and chronic wounds [5]. This explains the wound-healing efficacy of some EOs with anti-inflammatory and antioxidant activity [6,7,8]. In addition, in diabetic wounds, an important role is played by excessive ROS production or insufficient ROS detoxification related to hyperglycemia. In this scenario, antioxidant agents such as polyphenols have been suggested for diabetic wound treatment [9,10]. High oxidative stress is typical of burns, in which antioxidant therapy can be therefore promising [11], and even in corrosive esophageal burns, the use of antioxidants has been found to be effective at reducing inflammation and facilitating tissue healing [12].

CO is moreover characterized by analgesic and anesthetic effects. This last one, which has been largely exploited, especially in dentistry, can positively contribute to pain relief in wound treatment [2]. Anti-infective activity was also demonstrated towards Gram-positive (*Staphylococcus aureus*) and Gram-negative (*Escherichia coli*) bacteria, and towards *Candida albicans* for CO [13,14] as such, but also when formulated in chitosan nanoparticles (NPs) [15]. Eugenol antimicrobial activity was also pointed out by Antunes et al. [16] for CO and cinnamon leaf oil, both containing eugenol, loaded into chitosan and PVA films. The presence of EOs also increased the antibacterial activity observed for the unloaded chitosan against *Staphylococcus aureus* and *Pseudomonas aeruginosa*, microorganisms that often infect diabetic ulcers [16].

As with all the EOs, CO also requires suitable formulations to be administered in a moist environment, such as that of wounds. Some works have investigated the formulation of CO in nanofibers, with a positive effect on wound reduction due to the presence of the EO [1,17].

Besides nanofibers, nanoemulsions (NEs) also present some advantages, such as physical stability, an enhanced delivery efficacy, and protection against EO degradation and volatility [18]. Only a few studies until now have involved the use of CO NEs for wound healing. A pioneering article compared the healing effect after an oral administration of free CO and CO formulated into an NE, with gentamycin as a reference substance, in an excision wound rat model [19], finding a faster epithelialization process with both the CO and CO NE compared to the control.

In a further study, eugenol was formulated in an NE and the wound-healing effect was evaluated in an excision wound rat model through a topical application in comparison with pure eugenol (standard), with a faster wound closure observed after the treatment with the NE with respect to the control and standard eugenol [20]. Pereira dos Santos et al., based on the good antimicrobial activity of CO, designed a chitosan film loaded with a CO NE, with a potential application for wound healing [13]. Most recently, a CO emulsion based on C20/22 alkyl polyglucoside as an emulsifier, capable of forming liquid crystal structures, was tested in rat incision and excision wound models. The healing process was followed by a comparison with neomycin cream as a positive control. A statistically significant faster reduction in the wound area was observed with respect to the control for the CO emulsion. The peculiar structure due to the emulsifier was suggested as a relevant element for the healing improvement, as it contributed to wound hydration and protection [21]. 

The combination of a bioactive polysaccharide material with a hydrophobic moiety to obtain an amphiphilic structure suitable for the physical stabilization of NEs has been previously exploited in the case of chitosan oleate (CSO), a hydrophobically modified chitosan obtained through an ionic interaction with oleic acid [22]. Peculiar advantages can be derived from the maintenance in an NE of some biological activities of the polymer. It was demonstrated that CSO conserved the antimicrobial activity of both chitosan and oleic acid [23,24] and supported the antimicrobial activity of lemongrass EO towards bacterial strains with a synergic effect, improving the efficacy/toxicity ratio of the EO. In other cases, the combination of chitosan and EOs increased the antibacterial effect [22,25,26]. Moreover, for chitosan, some positive effects on wound healing have been described in the literature [27], and also, in the case of oleic acid, the evidence of its ability to activate pro-inflammatory effects and reduce necrotic tissue in wounds by increasing the neutrophil recruitment has been reported [28]. Their combination in CSO was proven suitable for the stabilization of *α*-tocopherol NPs [29]. These were positively associated with silver sulphadiazine in dressings that demonstrated a good efficacy in wound healing [30], supporting the relevance of antioxidant agents in tissue repair.

To better highlight these aspects, in the present work, we developed a CSO NE loaded with CO EO (CSO–CO NE), and its effect on ROS reduction and wound healing was studied. CSO was used for the first time as a stabilizer of a CO NE obtained by spontaneous emulsification, and their physical stability was then evaluated. The antioxidant activity of the CSO–CO NE was evaluated both in vitro by a 2,2-diphenil-2-picrylhydrazyl hydrate (DPPH) assay and in fibroblast cell lines by electron paramagnetic resonance (EPR) using *α*-phenyl-tert-butyl-nitrone (PBN) as a spin trap. The use of this test to detect the occurrence of ROS in cell lines has not been commonly reported in the literature, and one aim of the present work was to highlight its suitability for assessing the in vitro behavior of molecules and formulations. A murine burn model was used to evaluate macroscopic wound closure, and the skin reparation was evidenced through a histological analysis 18 days after the application of the developed CSO–CO NE. The literature does not describe, to our knowledge, the use of a burn model for the evaluation of the CO effect in wound healing; this approach could be useful to put in evidence the relevance of antioxidant properties on tissue repair. At 7 days from the injury, a biopsy of the wound was subjected to a histological analysis and a parallel dosage of lipid peroxidation by means of a TBARS assay. For this evaluation, as with the in vitro DPPH assay and ROS occurrence in fibroblasts, a direct comparison of the CSO–CO NE with the *α*-tocopherol treatment was performed, which is a feature that was not present for these aspects, to our knowledge, in previous literature works.

## 2. Materials and Methods

### 2.1. Materials

Acetonitrile (CHROMASOLV for HPLC gradient grade), oleic acid, calcium chloride, chitosan (low molecular weight, deacetylation rate 80%), clove oil (lot number MKBQ4410V), dimethyl sulfoxide (DMSO), 2,2-diphenyl-2-picrylhydrazyl hydrate (DPPH), Hank’s balanced salt solution (HBSS), Dulbecco’s modified Eagle’s medium (DMEM), fetal bovine serum (FBS), penicillin, streptomycin, amphotericin, thiazolyl blue tetrazolium bromide (MTT), porcine gastric mucin, *α*-fenil-*N*-tert-butyl-nitrone (PBN), Dulbecco’s phosphate-buffered saline (PBS), trypan blue, and trypsin-EDTA, 5-bromo-2′-deoxyuridine (BrdU) were purchased from Sigma Aldrich (Milan, Italy). A chemical characterization of a clove oil sample from the same supplier has been reported in the literature [1]. Acetone, ethanol, methanol, hydrogen peroxide solution, potassium chloride, monobasic sodium phosphate, and bicarbonate sodium were obtained from Carlo Erba (Milan, Italy). Baby normal human dermal fibroblasts (bNHDFs) were obtained from Promocell (Heidelberg, Germany), while Natrosol^®^ hydroxyethylcellulose 250 HX and HHX were a kind gift from Hercules S.p.A. Aqualon Division, Castel Maggiore, Bologna, Italy.

### 2.2. Preparation of Nanoemulsions

Chitosan oleate (CSO) salt was prepared through an ionic interaction between polymer-deacetylated amino groups and oleic acid carboxyl groups. Using this method, amphiphilic-modified chitosan occurs, able to spontaneously aggregate to form micelle structures that can be loaded with hydrophobic compounds.

Micelles of plain CSO were prepared using the solvent evaporation method: an oleic acid solution (1% *w*/*v* in ethanol) was added to a chitosan HCl solution (0.1% *w*/*v* in bi-distilled water) (molar ratio of 1:1 between deacetylated amino groups and carboxyl groups) [22]. Ethanol was then evaporated using nitrogen evaporation (1 h) and bi-distilled water was added to return the solution to the starting volume.

To prepare the NE clove oil (CO), the same procedure used to obtain plain CSO micelles was used, but the EO was solubilized in an ethanol solution with oleic acid to obtain chitosan–oleate/clove oil NE (CSO–CO NE). Two different formulations were prepared by varying the ratio between CSO and CO (CSO:CO), which was set to 2:1 or 1:1, corresponding to the theoretical loading of 50 and 100% *w*/*w* of the CO in the systems (CSO–CO 50 and CSO–CO 100, respectively) and the final CO concentrations of 1.2 and 2.4 mg/mL, respectively. Plain CSO at the same concentration as the NE was used as a control.

### 2.3. Particle Size Characterization

Photon correlation spectroscopy (PCS; N5 submicron particle size analyzer, Beckman Coulter, Milan, Italy) was used to determine the mean particle size and the polydispersity index (PI) of the NE. Each sample was diluted in filtered (0.22 µm) distilled water according to the apparatus instructions and the analyses were carried out at 25 °C and a 90° detection angle.

The NE physical stability was evaluated by preserving formulations at 4 °C for 6 months; at defined time points, the particle size distribution was evaluated to define the physical stability of all formulations.

### 2.4. Clove Oil Loading Evaluation

CO loaded in an NE was quantified by a spectrophotometric analysis (Perkin Elmer Instrument Lambda 25 UV/V is Spectrometer, Monza, Italy) at 280 nm. Briefly, for each formulation, 50 µL of the NE was diluted with an acetonitrile/acetate buffer (80:20 *v*/*v*) mixture to reach a final volume of 2 mL. The total CO content was evaluated from the standard calibration curve (r^2^ = 0.9997) obtained by analyzing a CO concentration range of 9–63 µg/mL.

The EO loading capacity (LC %) of each formulation was calculated as follows:$$LC\;\%=\frac{clove\;oil\;content}{chitosan\;oleate\;content+clove\;oil\;content\;}\times 100$$

The loading efficiency (LE %) was calculated as follows:$$LE\;\%=\frac{clove\;oil\;content}{clove\;oil\;used\;for\;nanoemulsion\;preparation\;}\times 100$$

### 2.5. Preparation of the Formulations Based on HEC

From the perspective of NE delivery to skin or mucosa, the CSO–CO 50, chosen as the model, was mixed at a 1:1 (*w*:*w*) ratio with hydroxyethyl cellulose (HEC) to obtain semi-solid formulations, suitable for topical applications. The HEC solutions were prepared in distilled water at concentrations of 1.5% and 3% (*w*/*w*) for HEC HX, and 1% and 1.5% (*w*/*w*) for HEC HHX. After complete polymer solubilization, they were mixed in a 1:1 (*w*/*w*) ratio with the CSO–CO 50 sample and left under magnetic stirring until complete homogenization was achieved.

### 2.6. Viscosity Determination of Nanoemulsions

All formulations were characterized in terms of their rheological properties. The viscosity measurements were carried out at three different temperatures, 25 °C, 32 °C (skin temperature), and 37 °C, using a rotational rheometer (Rheostress RS600, Haake, Karlsruhe, Germany) equipped with a cone plate combination C60/1 (diameter = 60 mm; angle = 1°), applying increasing shear rate values in the range of 10–300 s^−1^.

### 2.7. In Vitro Mucoadhesion Test

The formulations HEC HX 3%–CSO–CO 50 and HEC HHX 1.5%–CSO–CO 50 were subject to a tensile test, conducted using TA.XT plus a texture analyzer (ENCO, Spinea, Venice, Italy) equipped with a 1 Kg load cell and an A/MUC (mucoadhesion test ring) measurement system. The A/MUC measurement system consisted of a cylindrical substrate holder thermostated at 37 °C, into which a filter paper disc could be inserted. The filter paper was hydrated with 40 µL of 8% (*w*/*w*) gastric porcine mucin (type II, Sigma-Aldrich, Milan, Italy) or with 40 µL of simulated saliva in the case of blank measurements. Simulated saliva was prepared with NaHCO_3_ (0.42 mg/mL), NaH_2_PO_4_·H_2_O (0.91 mg/mL), NaCl (0.43 mg/mL), KCl (1.49 mg/mL), and CaCl_2_ (0.22 mg/mL) [31]. A preload force of 2.5 N was applied for 180 s. The probe was raised at 2.5 mm s^−1^ until the complete separation of the interface. Three replicates were performed for each sample.

The force (mN) required to detach the probe from the substrate was measured and the work of adhesion A (mN.mm) was calculated as the area under the curve of the mucoadhesion force (Fmax) vs. distance (mm).

In parallel, the contribution of HEC to mucoadhesion was evaluated by preparing two solutions of the polymer, HEC HX 3% (*w*/*w*) and HEC HHX 1.5% (*w*/*w*), and diluting them in a ratio of 1:1 (*w*/*w*) with bi-distilled water.

### 2.8. ROS-Scavenging Activity Assay

The ROS-scavenging activity of CSO–CO 50 and CSO–CO 100 NE was evaluated using the DPPH assay. Briefly, CSO–CO 50 and CSO–CO 100 NE were diluted with bi-distilled water to obtain a final CO concentration of 0.17 and 0.3 mg/mL, respectively. A total of 50 µL of each sample was mixed with 1950 µL of DPPH methanolic solution (6 × 10^−5^ M); the obtained mixtures were incubated and protected from light for 60 min and then spectrophotometrically analyzed at 515 nm (Perkin-Elmer Instrument Lambda 25 UV/Vis Spectrometer, Monza Brianza, Italy). *α*-tocopherol was considered a positive control, while as a negative control, a reaction mixture composed of water and DPPH solution in the ratio of 1:40 (*v*/*v*) was analyzed.

The percentage of ROS scavenging activity was calculated according to the following equation:$$DPPH^{•}\;scavenging\;effect\;\left(\%\right)=\left(1-\frac{A_{c}}{A_{s}}\right)\times 100$$
where *A_c_* is the absorbance of the negative control and *A_s_* is the absorbance of each sample.

The analyses were performed in three replicates.

### 2.9. In Vitro Cytotoxicity Evaluation

The cytotoxicity of all formulations was evaluated on baby normal human dermal fibroblasts (bNHDFs, Promocell, Heidelberg, Germany) cultured with Dulbecco’s modified Eagle’s medium (DMEM) enriched in 10% fetal bovine serum (FBS), penicillin (100 IU/mL), streptomycin (100 μg/mL), and amphotericin B (2.5 μg/mL). Cells were seeded in a 96-well plate (35,000 cells/well) and then treated with the chitosan derivative (CSO) and with the two NE formulations (CSO–CO 50, CSO–CO 100) diluted with culture medium (1:10, 1:20, and 1:40 (*v*/*v*) dilution rate). Untreated cells were considered controls.

After 3 and 24 h of incubation, at 37 °C and 5% CO_2_, all samples were aspirated and the cells were washed with 100 µL of Hank’s balanced salt solution (HBSS).

Then, 50 µL of thiazolyl blue tetrazolium bromide (MTT) solution was added to each well; viable cells, assessed by mitochondrial activity, metabolized the monotetrazolium salt in water-insoluble formazan crystals. After 3 h of incubation, the MTT solution was discarded and 100 µL of DMSO was added to induce formazan crystal solubilization. The optical density (OD) of the so-obtained solution was measured using an iMark™ microplate reader (Bio-Rad Laboratories S.r.l., Milan, Italy) at 570 nm and 670 nm (reference wavelength).

Cell viability (%) was calculated as: 100 × (OD treated cells/OD control cells). Untreated cells were considered controls.

### 2.10. In Vitro Cytoprotective Effect against Oxidative Stress

bNHDFs were seeded in a 96-well plate (35,000 cells/well) and treated with CSO–CO 50 and CSO–CO 100 NE diluted with culture medium (dilutions of 1:10, 1:20, and 1:40 (*v*/*v*). After 3 h of incubation, the cells were washed with PBS and treated with hydrogen peroxide (H_2_O_2_, 1 mM and 1.5 mM) for 1 h. At the end of the incubation time, an MTT assay was performed to evaluate the cell’s metabolic activity. Untreated cells were considered controls. Each condition was tested in triplicate.

### 2.11. Intracellular ROS Evaluation by EPR Method

In a 12-well plate (Cellstar^®^ tissue culture plate, Greiner Bio-One, Cassina de’ Pecchi, Italy), 4 × 10^5^ cells/well were seeded and left to incubate for 24 h. The medium was aspirated and the antioxidant samples were introduced, consisting of either 1 mL of *α*-tocopherol in a CSO sample [29] at the final *α*-tocopherol concentration of 10 µM or 1 mL of CS-CO 100 diluted to 1:20 (*v*/*v*) with the medium. Both the samples were left in contact with the cells for 3 h. A negative control consisting of untreated cells (100% damage) was also evaluated. The samples were removed and the damage was induced with 1 mM H_2_O_2_ for 1 h. Each well was washed with 1 mL of PBS and then 1 mL of an 18 mg/mL *α*-phenyl-*N*-tert-butyl-nitrone (PBN) spin trap solution was added and left for 30 min. The PBN solution was removed, a further washing of the wells was carried out, and the cells were subject to trypsinization to detach them from the bottom of the plate. After centrifugation, the supernatant was eliminated and for each analysis, the pellet obtained by joining two wells was used, taken up with 200 µL of Hank’s balanced salt solution (HBSS). The analysis was conducted with an EPR spectrometer (BRUKER EMX/12 with an ER4102ST cavity and a window for UV/Vis radiation, Bruker, Milan, Italy).

### 2.12. In Vivo Efficacy on Rat Wound Model

All animal experiments were carried out in full compliance with the standard international ethical guidelines (European Communities Council Directive 86/609/EEC) and were approved by the Italian Health Ministry (D.lgs.vo 116/92). The study protocol was approved by the Local Institutional Ethics Committee of the University of Pavia for the use of animals. Three male rats (Wistar, 200–250 g) were anesthetized with equitensine (3 mL/kg) and shaved to remove all hair from the site of injury. Three full-thickness burns, having a circular diameter of 4 mm, were produced on the animal’s back by contact with a brass rod (105 °C for 40 s). The day after, three 6 mm full-thickness excisional wounds were outlined using a punch biopsy tool on each animal’s back. The wounds were photographed with a digital camera and treated either with 100 µL of CO samples diluted to 1:1 (*v*/*v*) with an HEC solution (HX 3% *w*/*w*) or with a physiological solution (100 µL). The wounds were covered with sterile gauze and the rats’ backs were wrapped with a surgery stretch (Safety, Monza Brianza, Italy). At prefixed times after treatment (3, 7, 10, and 14 days), the three lesions were treated with 100 µL of CSO–CO NE samples diluted to 1:1 (*v*/*v*) with an HEC solution (HX 3% *w*/*w*) or wetted with a saline solution (NaCl, 0.9% *w*/*v*). At the end of the treatments (18 days), the lesions were photographed and the animals were labeled in vivo with a sterile solution of 10 mg/mL 5-bromo-2′-deoxyuridine (BrdU) by an intraperitoneal injection (100 mg/kg). After 60 min, the animals were sacrificed, full-thickness biopsies were obtained, and an immunohistochemical analysis of the excised tissues was carried out.

### 2.13. Microscopic Analysis

The tissue samples were bisected along the widest line of the wound, then fixed in 4% *w*/*v* neutral-buffered paraformaldehyde for 48 h, dehydrated with a gradient alcohol series, cleared in xylene, and embedded in paraffin. Sections (8 μm) were obtained using a Leitz (Wetzlar, Germany) microtome and were stained with hematoxylin and eosin (H&E) or subjected to the immunohistochemical detection of BrdU. The slices were examined at a magnification of 5× under a light microscope, Axiophot Zeiss (Oberkochen, Germany), equipped with a digital camera (Sigma SD14, Sigma Mtrading s.r.l., Opera, Italy).

### 2.14. Immunohistochemical Detection of BrdU

Sections (8 µm) were deparaffinized in xylene and hydrated in a series of graded alcohols to water, and then slides were immersed in a blocking reagent (Biocare’s Peroxidazed 1 blocking reagent, Biocare, Yorba Linda, CA, USA) for 5 min at room temperature (RT) and subsequently washed in distilled water. For heat-mediated antigen retrieval, the slides were placed in Rodent Decloaker buffer (Biocare, Yorba Linda, CA, USA) and heated to 100 °C for 30 min by a steamer; when the slides became cold, Rodent Block M (Biocare, Yorba Linda, CA, USA) was applied for 30 min. After washing with tris-buffered saline (TBS), the primary antibody (monoclonal anti bromodeoxyuridine, Biocare, Yorba Linda, CA, USA) was applied in a ratio of 1:100 (*v*/*v*) for 2 h at RT. This antibody reacts with BrdU in single-stranded DNA, for BrdU attached to a protein carrier or free BrdU. It detects nucleated cells in the S-phase that have had BrdU incorporated into their DNA. The samples were subsequently washed in TBS and then covered with Mouse-on-Mouse HRP-Polymer (Biocare, Yorba Linda, CA, USA) for 20 min at RT. After thoroughly washing in TBS, the slides were incubated for 5 min with a chromogen solution (Biocare’s DAB, Biocare, Yorba Linda, CA, USA). Positive and negative controls were performed as well.

### 2.15. Lipidic Peroxidation Evaluation by Thiobarbituric Acid (TBA) Assay

Three more animals, subject to burn and excision wounds as described above, were each treated with a saline solution (NaCl, 0.9% *w*/*v*) as a positive control, with CSO–CO 100, or with *α*-tocopherol (0.33 mg/mL) in CSO micelles prepared as previously described [30], for comparison purposes. The rats were, in this case, sacrificed after 7 days of treatment, and a thiobarbituric acid-reactive substances (TBARS) measurement was conducted on full-thickness biopsies as described in the literature [32]. Briefly, tissues were homogenized in a KCl solution (1.15% *w*/*v*). An amount of 100 µL of each homogenized tissue was added to the reaction mixture (70 µL of 8.1% sodium dodecyl sulfate (SDS), 500 µL of 20% acetic acid with a pH of 3.5, 500 µL of 0.8% TBA, and 70 µL of distilled water). The obtained mixtures were boiled (1 h, 95 °C) and then centrifuged (3000× *g*, 5 min).

The supernatants were analyzed using a microplate reader (Synergy HT, BioTek, Swindon, UK) at 570 nm (OD570). Malondialdehyde solutions (0.05, 0.1, 0.25, and 0.5 mM) were prepared and analyzed with the same method to obtain a calibration curve (R^2^ = 0.9995).

Healthy cutaneous tissues were considered as a negative control.

### 2.16. Statistical Analysis

The statistical analyses of the data were carried out with a statistical software package (Statgraphics Centurion 16, Statistical Graphics Co., Rockvillle, MD, USA). Student’s *t*-test for independent variables was performed to compare the means of two sets of data. For the statistical analysis of multiple samples, one-way analysis of variance (1-way ANOVA) was performed, followed by a Fisher’s least significant difference (LSD) procedure to evidence which means were significantly different from the others.

## 3. Results and Discussion

### 3.1. Particle Size Characterization of Nanoemulsions

Table 1 reports the results of the dimensional characterization, given by the mean diameter from the number and volume distributions and the polydispersion index (PI) for the NE obtained with CSO at two different CO ratios (CSO–CO 50 and CSO–CO 100), corresponding to final CO concentrations of 1.2 and 2.4 mg/mL, respectively. The NEs were prepared by spontaneous emulsification using ethanol as the organic phase. It must be noted that ethanol is one of the most acceptable solvents, considering not just health and safety concerns, but also the environmental impact, from the perspective of a green industrial approach [33].

In both formulations, the dimensions were quite small, with a mean diameter between 200 and 250 nm, and the PI, remaining below 0.5, was compatible with a homogeneous distribution independently of the CO concentration tested. It must be noted that the use of volume distribution resulted in mean diameters higher than those obtained using a number distribution. This last factor resulted in values compatible with the strict definition of a nanomaterial (at least 50% of the distribution lower than 100 nm). The volume distribution was selected as an identifying parameter due to its better ability to put in evidence instability phenomena such as aggregation or coalescence.

The NEs with both CO concentrations have been studied in terms of their physical stability by maintaining them at 4 °C for 6 months, and the results obtained over different times of storage are illustrated in Figure 1. For both NEs, after an initial slight increase in dimensions, a stabilization was observed with dimensions of around 300 nm during the remaining period of storage. In addition, the PI showed an initial increase followed by a stabilization at values mainly below 0.5, indicating an acceptable dimensional dispersion and the ability of CSO to stabilize the systems.

Because the hydrophobically modified chitosan, used here as the only stabilizer of the NEs, was obtained by an ionic interaction, to assess that its performance was not affected by a pH variation and/or by the medium ionic strength, the physical stability of NEs was also carried out in a slightly acidic solution (acetate buffer, pH of 4.0) and a saline solution (0.9% (*w*/*v*) NaCl), and the effects of these two different conditions on the dimensions and PI (Figure 2) were also verified.

Both NE formulations appeared unchanged in the saline solution for up to 1 month, while in the acidic medium, only a slight increase in dimensions was observed, with the PI values in all cases below 0.5. This suggests, despite the ionic character of the polymer derivative, a suitable physical stability of the NEs for their potential application in a complex physiological environment.

Many reasons encourage the research of new emulsifiers, among which a growing interest is in natural emulsifiers. These meet the increasing request for all-natural formulations, and from this perspective, biopolymers such as proteins and polysaccharides offer a wide range of choices [34,35]. All emulsifiers act at the oil–water interface by decreasing the interfacial tension, and quite often, the stabilization by macromolecules relies on the steric effect that follows adsorption at the droplet surface and impairs flocculation, coalescence, and Ostwald ripening [36]. Polysaccharides are quite efficient in this mechanism thanks to their large molecular weight, and they are less susceptible to changes in environmental pH and ionic strength than proteins. However, among the limits of polysaccharide-based surfactants is that they are not suitable for low-energy emulsification methods such as self-emulsification [36]. The self-assembly of chitosan and oleic acid to form the amphiphilic chitosan derivative CSO in situ represents an exception, and the NE is formed and stabilized under mild stirring without any homogenization step. This can represent an advantage, especially for the formulation of substances such as EOs, which are easily subject to degradation and evaporation at the high temperatures that develop during high shear homogenization [37]. The thickness of the shell that forms around the droplets by macromolecule adsorption can moreover protect volatile EOs from loss more efficiently than a thin layer of a small-molecule surfactant.

### 3.2. Clove Oil Loading in Nanoemulsions

Figure 3 shows the CO loading in the obtained NEs, expressed as the loading yield efficiency, which indicates the percentage of CO found in the final emulsion with respect to the theoretical one and the loading capacity %, which takes into account the presence of the stabilizer CSO. For both the samples, some decrease with respect to the theoretical concentrations was observed; in particular, the theoretical values decreased from 1.2 and 2.4 mg/mL to 0.85 ± 0.04 and 1.74 ± 0.12 mg/mL in the CSO–CO 50 and CSO–CO 100 systems, respectively, with a loss lower than 30%. This result seems to be in accordance with the results obtained for the encapsulation of eugenol in other systems, such as in chitosomes [38].

Considering the high volatility of EOs, the observed decrease was conceivably due to the CO loss that occurred during the preparation of the systems, especially in the most critical steps, such as EO/ethanol dropping and the subsequent removal of the organic phase by evaporation.

In Figure 4, the amount of CO quantified in the NEs is reported during 4 months of storage in a refrigerator (4 °C) after the preparation. With respect to the initial amount, considered here to be 100%, a decrease could be observed for both samples in the first 4 weeks; after that, the CO value became quite stable, confirming the stabilizing effect of the CSO NE for volatile substances such as EOs.

### 3.3. Viscosity and Mucoadhesion Properties of Nanoemulsions

From the perspective of NE delivery to skin or mucosa, the CSO–CO 50 NE, chosen as the model, was mixed in a ratio of 1:1 (*w*/*w*) with hydroxyethyl cellulose (HEC) to obtain semisolid formulations, which were suitable to be applied using spray or drop devices. HEC is an easily soluble viscosity modifier that was chosen because, for its non-ionic nature, it seemed less prone to react with both chitosan and oleic acid. In Figure 5, the results of the rheological characterization of the 1:1 (*w*/*w*) mixtures with high (HHX)- or low (HX)-molecular-weight HEC are given. Solutions of each molecular weight of HEC were tested at two concentrations, 1.5% and 3% (*w*/*w*) for HX and 1% and 1.5% (*w*/*w*) for HHX, to modulate the viscosity. All the formulations were tested at 25 °C and 32 °C (Figure 5a,b) to evaluate their behavior at simulated room and skin temperature. It was possible to identify two more viscous and markedly pseudoplastic formulations (HX 3% and HHX 1.5%) suitable to be delivered by a drop device (VP6 dropper, Aptar Mezzovico SA, Mezzovico, Switzerland). The formulation with the lower-viscosity HX (1.5% *w*/*w*) seemed suitable to be delivered by a spray device (VP6 dropper, Aptar Mezzovico SA, Mezzovico, Switzerland). The variability in the amount of the tested formulations delivered from the dropper and the spray VP6 devices (the latter in the case of the low-viscosity HX 1.5%) is illustrated in Figure 5c. The two most viscous formulations (HHX 1.5% and HX 3%) were also tested after 1 and 2 weeks of storage (T1 and T2). In all cases, the quite-low standard deviation indicated a homogeneous dispensed amount obtained with both the devices, and for HHX 1.5% and HX 3%, it also indicated quite a good physical stability during the tested times. The viscosity of the two most viscous formulations was also tested at 37 °C (Figure 5d), envisaging an application to mucosal lesions. The possible application of the CSO–CO NE for the treatment of mouth herpetic stomatitis lesions could also be suggested by the antiviral activity reported in the literature for eugenol [39]. For a possible mucosal administration, for the two formulations, HX 3% and HHX 1.5%, a mucoadhesion test was also performed, the results of which are illustrated in Figure 6.

For both samples and both mucoadhesion parameters considered (Fmax and adhesion work), it was observed that all the results obtained with the blanks (without mucin) were comparable. All the formulations except the HEC HHX 1.5% without the CO NE showed a statistically significant positive interaction with the mucin compared to the blank (without mucin), both in terms of maximal mucoadhesion force and adhesion work (Student’s *t*-test, *p* < 0.05). In the case of the HEC HX 3% sample, for which even the polymeric solution alone was mucoadhesive, a more marked increase in the interaction with the mucin was obtained after mixing with the NE, confirming the ability of the chitosan shell of the droplets to maintain the known mucoadhesive property. The Fmax increase was 22.78% and 23.73%, respectively, for the formulations with HEC HX 3% and HEC HHX 1.5%, while the work performed was higher by 35.93% and 24.63%. It is possible that in the case of the 3% HX sample, the lower molecular weight corresponds to a higher polymer chain mobility and interactions with the mucin chains.

It is worth observing that in the present system, the mucoadhesion effect due to chitosan was localized at the surface of every single droplet. This allowed for better contact of the encapsulated oil with the mucosal surface, further optimized by the increased surface/volume ratio due to the nanometric dimensions of the droplets.

### 3.4. ROS-Scavenging Activity Evaluation

Figure 7 shows the antioxidant activity in the DPPH test of unloaded CSO, at the same concentration of the NE samples, and of CO, as such and encapsulated in a CSO NE, at two different concentrations: 0.17 and 0.36 mg/mL. A comparison with *α*-tocopherol at the same concentrations is reported too, chosen as a reference because of its well-known antioxidant activity.

It can be seen that the CO activity is comparable with that of *α*-tocopherol, and it is maintained even after encapsulation in an NE. There is no statistical difference, at the same concentration, between CO as such and encapsulated CO (Student’s *t*-test, *p* < 0.05).

### 3.5. Cytotoxicity Evaluation

In Figure 8, the results of the cytotoxicity test for the two NE samples are given after 3 h (Figure 8a) and 24 h of contact (Figure 8b). The same concentrations of CSO present in the NE samples were also tested under the same conditions and considered as a blank.

After 3 h of contact, for all samples, only the highest concentration corresponding to the 1:10 (*v*/*v*) dilution resulted in a significant decrease in cell viability compared to the control (ANOVA 1-way, Fisher’s LSD test, *p* < 0.05).

For CSO, both the samples diluted to 1:20 and 1:40 (*v*/*v*) were compatible with the cells over short and long contact times. After 24 h, an increase in viability due to the proliferative effect of CSO was seen, which is in line with previous evidence [40].

In the case of the CSO–CO 50 and CSO–CO 100 NE samples, the viability of the 1:40 (*v*/*v*) dilution was close to 100% up to 24 h, and not statistically different from that of the controls (ANOVA 1-way, Fisher’s LSD test, *p* < 0.05). The 1:20 (*v*/*v*) dilution resulted in biocompatibility after 3 h of contact, but at 24 h, it showed a decrease in viability that seemed to be due mainly to CO, as the corresponding CSO sample was still biocompatible.

### 3.6. EPR Results

The antioxidant effect of the CSO NE was also studied in fibroblast cell lines, putting in evidence the ROS production by employing electron paramagnetic resonance (EPR). In this case, fibroblasts were treated with H_2_O_2_ at 1 mM and 1.5 mM, with and without a previous treatment with CSO and the CSO-loaded samples (CSO–CO 50 and CSO–CO 100) at a 1:20 (*v*/*v*) dilution, which was previously identified as safe for the cells for a 3 h exposure time (Figure 8). An MTT test was carried out to determine the cell viability in the presence of H_2_O_2_ (Figure 9).

All the cells, even those treated with H_2_O_2_ only, maintained a high level of viability. The cells treated with H_2_O_2_ (1 mM) with and without a previous treatment with CSO–CO 100 NE (1:20 *v*/*v*) were studied for ROS presence using the EPR test. Fibroblasts subject to the same treatment with H_2_O_2_ and with 3 h of protective treatment with *α*-tocopherol encapsulated in CSO at a concentration of 10 µM, as previously described and characterized [29], were tested under the same conditions for comparison purposes.

The EPR signal intensity was linearly correlated with the number of trapped radicals. The results are expressed in arbitrary units, normalized depending on the number of scans (which might differ from sample to sample).

The results are illustrated in Figure 10. The signal in Figure 10a, concerning fibroblasts without protective treatment, is clearly higher (16,429/scans) than that of Figure 10b,c (which have enlarged y-axes), corresponding to a signal of 923/scans and lower than 250/scans, respectively. The protective effect of the CO NE appears to be very clear in this case, close to that of *α*-tocopherol thanks to the higher sensitivity of the ROS quantification method used, which is capable of detecting high ROS levels before the damage of the cells becomes high enough to lead them to death.

The term ROS describes a group of all the chemical species that contain oxygen, coming from the incomplete reduction of oxygen, among which are various free radicals such as superoxide radicals and hydroxyl radicals, but also including non-radical oxidizing agents. They are highly reactive species and cause damage to nucleic acids and proteins, leading to cell death. The high reactivity is also the reason for the short lifetime of ROS, explaining why it is quite difficult to detect them inside cells. EPR is the main technique used to detect ROS and radical species, but the use of redox probes or spin traps is required to detect the transient ones. These probes interact with the ROS present inside the cell, forming paramagnetic compounds that are stable for long times, and therefore detectable by EPR [41].

The EPR spin probe technique has been proposed in vivo in rat models for evaluating ROS generation and decay in different lesion models [42]. The use of the method specifically in cell cultures has also been proposed [43].

Although qualitative, the results obtained here show that the use of EPR with a PBN probe is effective for the detection of ROS inside cells after oxidative stress, such as that caused by H_2_O_2_ here, even when their levels are still too low to cause cell death. Fibroblasts exposed to H_2_O_2_ showed 80% viability, but a clear signal was visible in EPR (Figure 10a). Quite clear results in these conditions were also given by the reduction in the signal observed in cells exposed to H_2_O_2_ with the protection given by the two antioxidant samples (Figure 10b,c), which were maintained in contact with the cells for 3 h before their removal and the addition of H_2_O_2_. These findings also confirmed the high sensitivity of the EPR technique, combined with the use of a suitable spin trap, to detect and quantify radical species [44].

### 3.7. In Vivo Study—Macroscopic Evaluation, Histological Examination, and Immunohistochemical Detection of BrdU

The images obtained for the wounds at the time of excision, that is, 24 h after the thermal injury, are reported in Figure 11a,c,e. The images of the same wounds after 18 days of the different treatments are given in the right part of the figure, in Figure 11b,d,f. The quite fast closure of the wound treated with the sample CSO–CO 100 was observed (Figure 11d). The CSO–CO 50 effect (Figure 11b) seemed to be intermediate between that of the CO at the highest concentration and of the saline solution (Figure 11f), suggesting, although qualitatively, a dose-dependent response.

Looking at the histological results illustrated in Figure 12, after 18 days of treatment with CSO–CO 50 (Figure 12a,b), the epidermal layer was not yet fully restored, and necrotic material was detectable over the lesion. The underlying connective tissue contained just a fair inflammatory infiltrate; granulation tissue as well as dilated vessels were present in a limited area just beneath the not yet re-epithelized part, while hereinafter, collagen fibers were already organized in large bundles typical of the reticular layer of the dermis. On the contrary, in the saline-treated samples, the underlying connective tissue was affected by an abundant inflammatory infiltrate with numerous dilated vessels, while the collagen fibers were dispersed and not yet organized. In the CSO–CO 100-treated specimens (Figure 12c,d), on the other side, the epidermal layer appeared thick and well organized in several cell layers and showed a fair degree of keratinization (cornification). Inflammatory infiltrate and abnormally dilated vessels were almost completely absent. These results are in accordance with other published data; in particular, Aman and colleagues demonstrated that the vehiculation of CO in nano-sized emulsions facilitates the cellular uptake of EOs by the cells involved in inflammatory reactions [45]. A fine network of collagen fibers was restored at the dermo-epidermal junction and the bundles of collagen fibers in the dermis had a normal size. The hair follicles were reforming. Although the samples treated with CSO–CO 50 were in a more advanced stage of wound healing compared to the saline-treated ones (Figure 12e,f), in which the part that had not re-epithelized appeared on average to be more extended in all the analyzed samples, the features of the CSO–CO 100-treated samples better resembled a deeply intact skin picture (Figure 12g,h), displaying a well-organized epidermis in several cell layers and a high degree of keratinization. The underlying connective tissue had large bundles of collagen fibers, typical of the reticular layer of the dermis, as observed in the physiological assessment. In fact, skin appendages such as hair follicles and glands were detectable and well defined. To confirm that the samples treated with CSO–CO were in a more advanced stage of wound healing, the proliferative activity in the treated samples—pointed out by nuclear staining for BrdU—appeared less developed than that in the saline solution-treated specimens, where labeled nuclei were more present both in the dermis and epidermis layers. The saline solution controls, therefore, could be considered to be in the “proliferative phase”, a more rearward phase compared to the “remodeling phase”, presumably in which were the CSO–CO-treated specimens.

These results, therefore, confirm what was previously observed in the literature by Banerjee [21], although a direct comparison is made difficult by the different wound models, which in the present case involved an initial burn, likely to cause a strong inflammatory response and the generation of high ROS levels. To put in evidence the role of these aspects in lesion healing, three more rats were subject to the same burn and excision treatment, but were sacrificed after only 7 days and two treatments with the NE samples (at the time of excision, and after 3 days). In this case, the comparison was performed between the CSO–CO 100 sample, which seemed the most promising, and a sample containing *α*-tocopherol at 0.33 mg/mL encapsulated in CSO according to Bonferoni et al. [29], to allow for the comparison of the CO NE with a well-known antioxidant treatment.

In Figure 13, it is possible to see the sections of healthy rat skin (Figure 13a) and of the wounds after 1 week of treatment with the saline solution (Figure 13b), with *α*-tocopherol (Figure 13c), or with CSO–CO 100 (Figure 13d), with 100 µL of each. In the healthy skin, the pluristratified and cornified epithelium completely covered the surface, the dermis was organized in large acidophilic bundles, and numerous skin appendages were evident (hair follicles and sebaceous glands). It can be seen that in the lesion with the saline solution, the re-epithelialization process was not yet appreciable on the injured part, from which the necrotic material partially detached; the dermis was rich in inflammatory cells—in which the basophil nuclei were visible—and in the blood vessels, some of which appeared very dilated. In the case of treatment with *α*-tocopherol, the surface of the lesion, not yet re-epithelialized, was covered by acidophilic necrotic material, while the part immediately below the area infiltrated by inflammatory cells—in which the basophilic nuclei could be seen—was thin. However, numerous blood vessels were evident, some of which were dilated. In the wound treated with CSO–CO 100 (Figure 13d), the re-epithelialization process was not yet appreciable on the injured part, from which the necrotic material partially detached; no inflammatory cells were seen in the dermis, even though numerous blood vessels were present.

### 3.8. Thiobarbituric Acid (TBA) Assay Results

To verify the relationship between the inflammatory conditions of the tissues and the level of lipid peroxidation, and to highlight how the lower inflammation detected in the presence of the formulations with *α*-tocopherol and CSO–CO 100 could be related to their antioxidant action, a method for the quantification of thiobarbituric acid-reactive substances (TBARS), products of peroxidation, was applied to wound biopsies taken at 7 days [32]. Based on a calibration curve using malondialdehyde as the standard, the TBARS present in rat skin samples were determined: samples of healthy skin and samples of burnt skin after 7 days of treatment with a saline solution, *α*-tocopherol (0.33 mg/mL) in CSO, or CSO–CO 100 (Figure 14). A statistical analysis (ANOVA 1-way, Fisher’s LSD test, *p* < 0.05) showed that the TBARS levels of both the CSO–CO 100 and *α*-tocopherol samples were statistically different from those of healthy skin and the saline solution samples, while no statistically significant difference was found between them. The burnt plug removed at T0 was also assessed for TBARS content, and a value of 1.35 ± 0.17 nmol/mg skin was found, indicating a high level of peroxidation induced by the thermal lesion. After 1 week, even with the saline solution treatment, the TBARS level was reduced, although it appeared clear that both the treatments with antioxidants (*α*-tocopherol and CO) reduced the peroxidation levels faster, making them closer to those of the healthy skin after just 1 week of treatment.

The formulation of anti-inflammatory and antioxidant EOs in NEs can further support their efficacy thanks to the high surface-to-volume ratio of the droplets, improving the interaction with the biological substrate. A further advantage can be derived from natural surfactants with peculiar biological activities that are able to support the therapeutic application of the oil phase. An example is given in the present case by CSO.

Chitosan is well known for having a lot of effects that are useful during wound healing, such as the previously mentioned anti-infective properties, and some antioxidant effects have been found for chitosan with a high deacetylation degree (DDA) and a low molecular weight (MW) [46,47]. Anti-inflammatory activity has been studied for chitosan by many authors, and a reduction in pro-inflammatory cytokines has been observed, especially with low-molecular-weight samples [46]. Properties such as the MW and DDA can also strongly affect the pro-inflammatory or anti-inflammatory response to chitosan, contributing to the complex scenario of results regarding its role in cytokine induction and macrophage activation and polarization [48]. Similarly, studies investigating the effect of oleic acid on inflammation have put in evidence an anti-inflammatory effect induced by an increase in IL10 levels and a reduction in COX-2 levels [28]. This is in line with the improvement of skin wound healing that was observed in rats following supplementation with fatty acids [49] and with findings of topical oleic acid anti-inflammatory activity [50]. However, other authors have found a high expression of pro-inflammatory TNF and IL-17 in oleic acid-treated wounds [28]. This result is supported by Pereira et al. [51], who found that oleic acid induced proinflammatory effects on wound healing, the stimulation of neutrophil migration to the lesions, and the release of VEGF and IL-1. These different results can be interpreted by taking into account that inflammation in the early phase is important to activate the processes that lead to wound healing. In the case of chitosan and oleic acid, therefore, a complex balance of pro- and anti-inflammatory activities plays a role in the modulation of repair processes, explaining the positive effect that is observed in the literature and in the present study for their use in wound healing.

## 4. Conclusions

The stabilization of CO with CSO results in low dimensions and a physically stable NE with good protection of the volatile EO encapsulated, conceivably due to the thick shell of polysaccharides around each EO droplet.

The low dimensions of the dispersed phase improve the interaction with the biological substrate, allowing the efficient spreading of the EO. For the application of NEs to skin wounds, the effect due to the contact of the oil with the wet wound area is moreover potentially supported by the antimicrobial, anti-inflammatory, and wound-healing favoring effect of chitosan and oleic acid. In the case of mucosal lesions, further advantages can arise from chitosan’s mucoadhesion properties, which allows the prolonged contact of the EO with the application site.

Although qualitative, the EPR analysis of ROS in cells treated with H_2_O_2_ confirms the protective effect of the CSO–CO NE, which was comparable to that obtained with the well-known antioxidant *α*-tocopherol. The combination of EPR with a PBN probe was confirmed as a highly sensitive method for the detection of ROS inside the cells after oxidative stress.

The in vivo results confirm the promising effect of CO in wound healing, as previously observed in the literature, even in the rat model used here, based on the thermal lesion and excisional wound. Inflammation and peroxidation reduction were quite clear in this model for the most promising CSO–CO 100 sample, the activity of which was comparable, even in this case, with that of *α*-tocopherol.

## Figures and Tables

**Figure 1 antioxidants-12-00273-f001:**
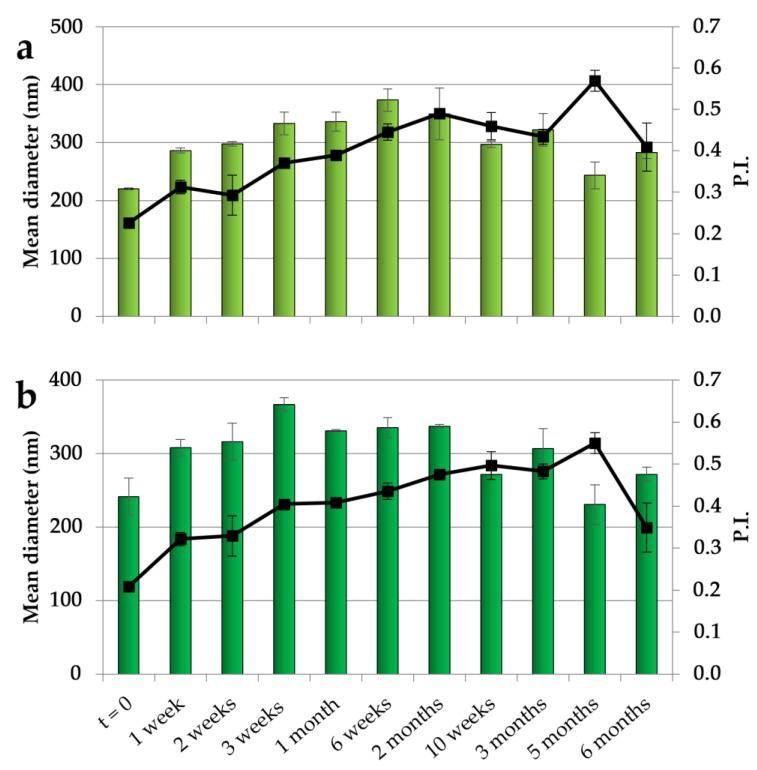
Dimensions (histogram bars) and PI (black symbols) of CSO–CO 50 (**a**) and CSO–CO 100 (**b**) NEs at different times of storage (4 °C) (mean ± sd; n = 3).

**Figure 2 antioxidants-12-00273-f002:**
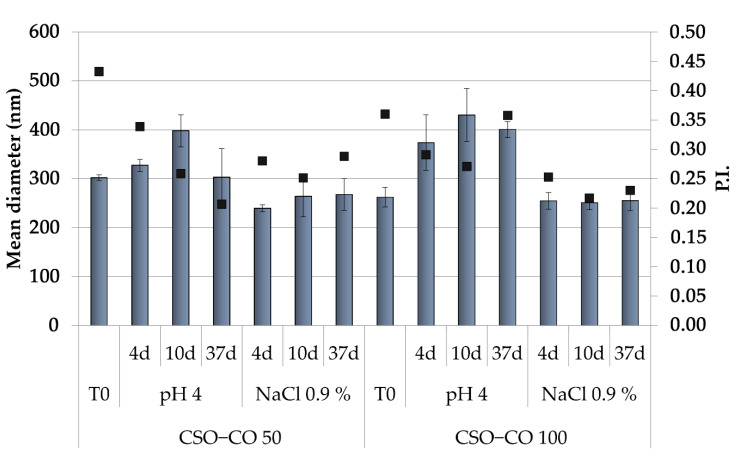
Dimensions (histogram bars) and PI (black symbols) of CSO–CO 50 and CSO–CO 100 maintained in acidic pH and saline solution (NaCl 0.9% *w*/*v*) after different times up to 37 days (mean ± sd; n = 3).

**Figure 3 antioxidants-12-00273-f003:**
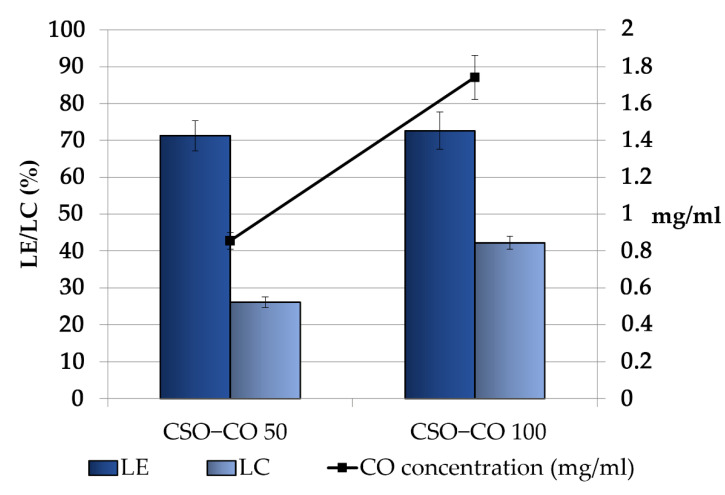
Loading efficiency (LE (%)), loading capacity (LC (%)), and final concentration of CO in the CSO–CO 50 and CSO–CO 100 NE samples just after the preparation (mean ± sd; n = 3).

**Figure 4 antioxidants-12-00273-f004:**
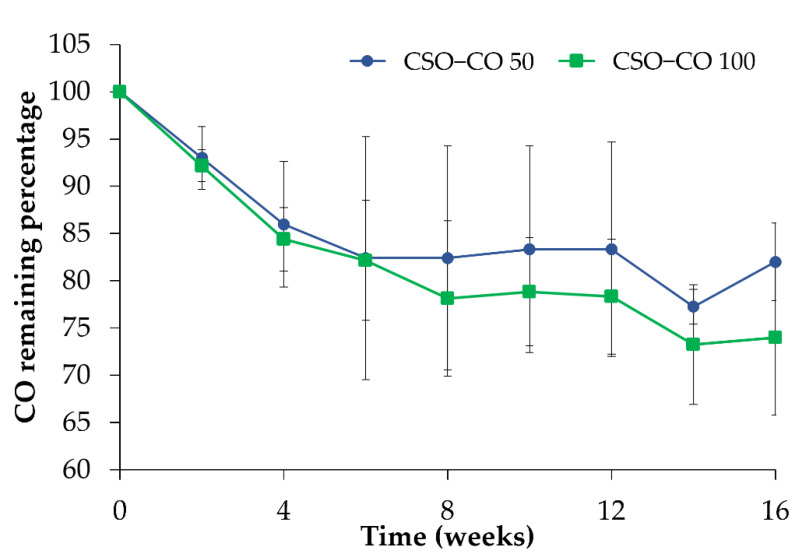
CO assay (%) in the CSO–CO 50 and CSO–CO 100 NE samples during storage at 4 °C (mean ± sd; n = 3).

**Figure 5 antioxidants-12-00273-f005:**
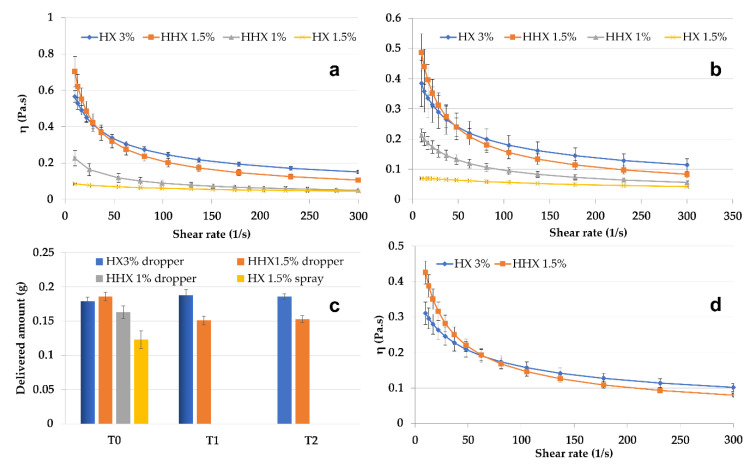
Viscosity curves of CSO–CO 50 diluted to 1:1 *w*/*w* with HEC low (HX) and high (HHX) molecular weights at different concentrations at 25 °C (**a**) and 32 °C (**b**) (mean ± sd; n = 3). Delivered amounts of HHX 1% and HX 1.5% through VP6 dropper and VP6 spray devices at T0 and after 1 week (T1) and 2 weeks (T2) (**c**) (mean ± sd; n = 10). Viscosity curves of the two most viscous samples, HX 3% and HHX 1.5%, at 37 °C (**d**) (mean ± sd; n = 3).

**Figure 6 antioxidants-12-00273-f006:**
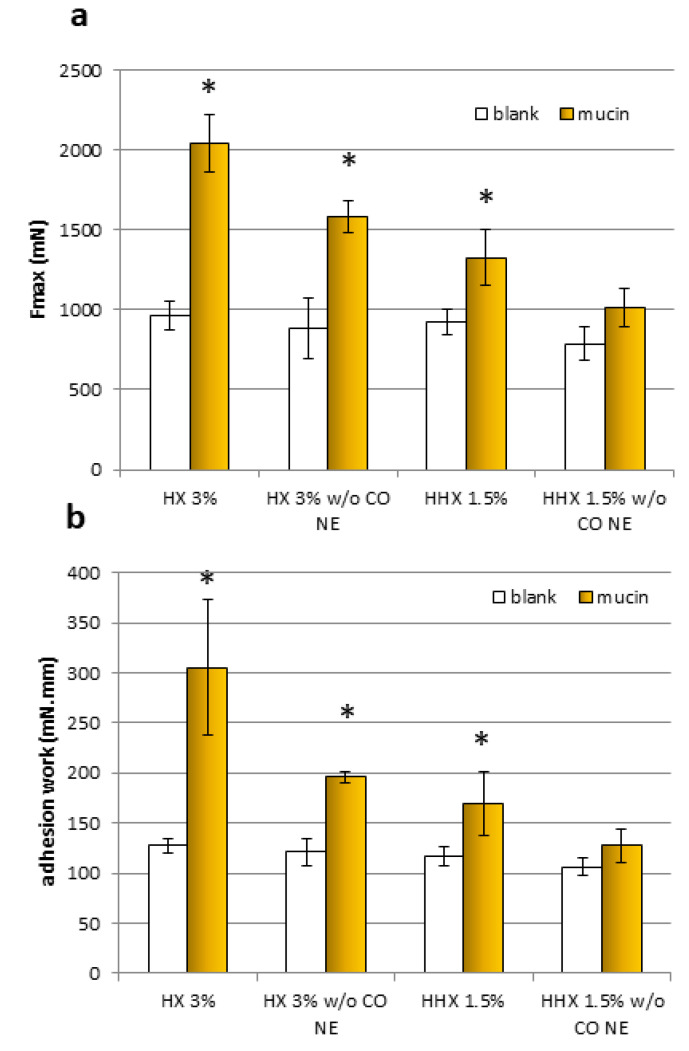
The maximum force of detachment, Fmax (**a**), and work of adhesion (**b**) for the samples, HX 3% and HHX 1.5%, and for the HEC alone at the same concentrations (*w*/*o* EO NE) (mean ± sd; n = 3). * Indicates samples statistically different from the corresponding blanks (Student’s *t*-test, *p* < 0.05).

**Figure 7 antioxidants-12-00273-f007:**
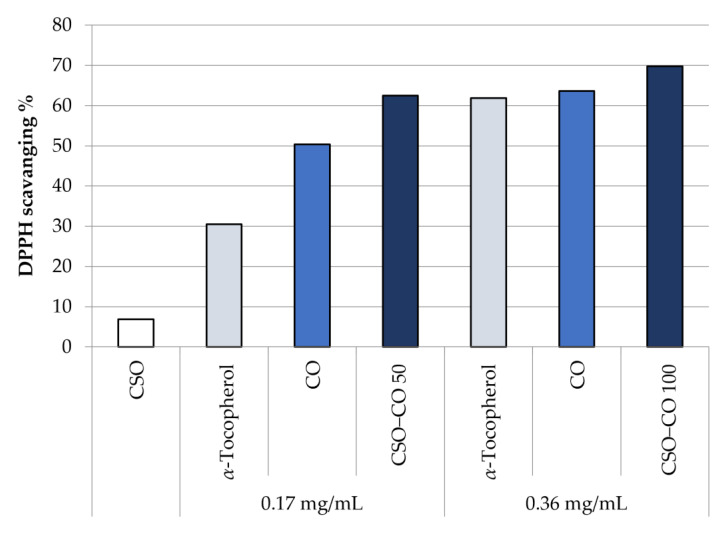
DPPH scavenging effect of the unloaded CSO NE, *α*-tocopherol, free clove oil (CO), and CO encapsulated in NEs at two concentrations (CSO–CO 50 and CSO–CO 100) (mean ± sd; n = 3).

**Figure 8 antioxidants-12-00273-f008:**
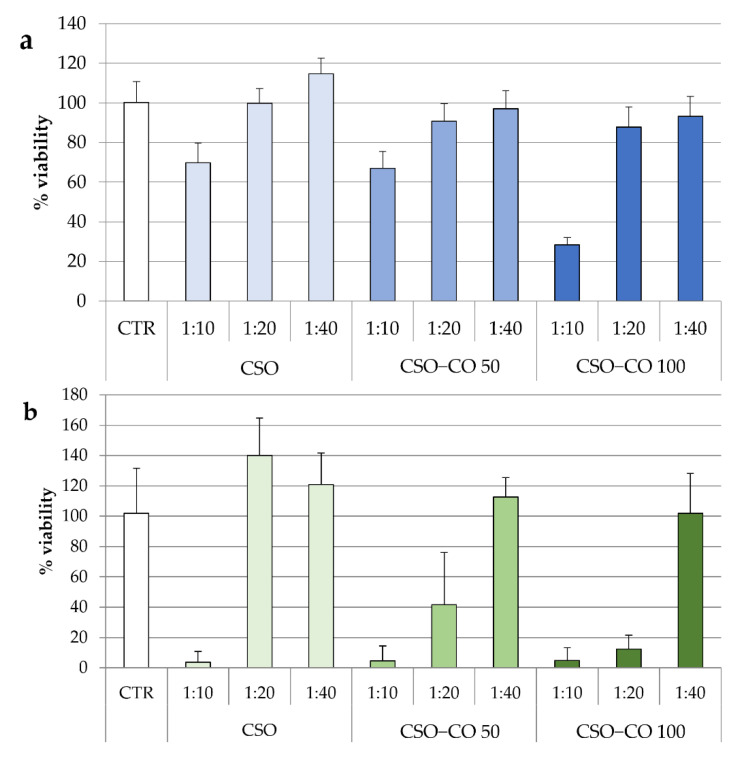
Fibroblast viability after 3 h (**a**) and 24 h (**b**) of treatment with unloaded CSO as a blank and with the CSO–CO 50 and CSO–CO 100 NE samples at different dilutions with culture medium (1:10, 1:20, and 1:40 *v*/*v*) (mean ± sd; n = 8).

**Figure 9 antioxidants-12-00273-f009:**
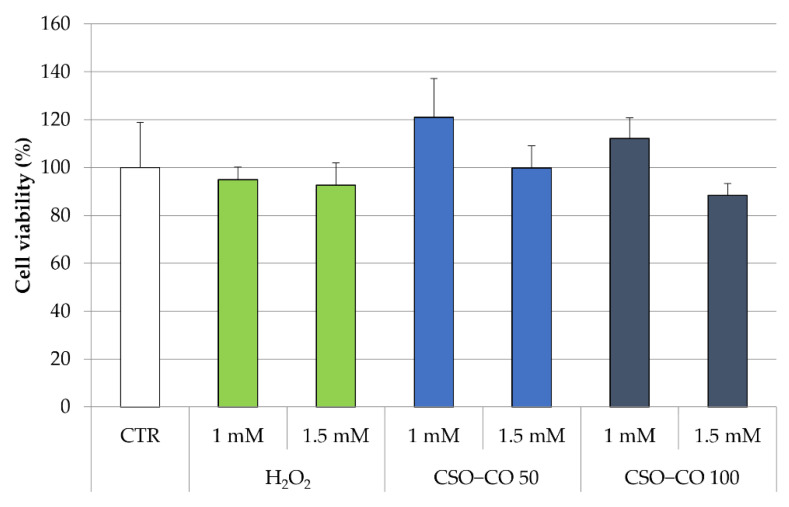
Cell viability (%) of fibroblasts treated with H_2_O_2_ with and without a previous treatment with CSO–CO 50 and CSO–CO 100, both diluted to 1:20 (*v*/*v*) (mean ± sd; n = 4).

**Figure 10 antioxidants-12-00273-f010:**
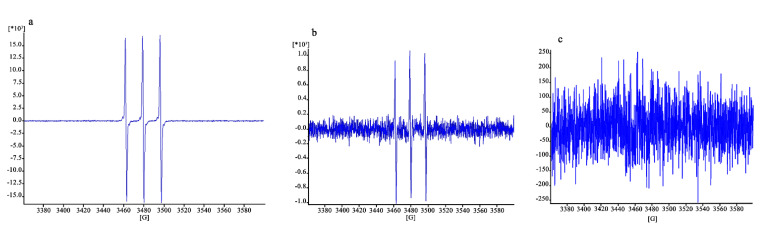
EPR signal due to the ROS inside the cells damaged after exposure to H_2_O_2_ without previous treatment (**a**) and with treatment with CSO–CO 100 (1:20 (*v*/*v*)) (**b**) or *α*-tocopherol (10 µM) encapsulated in CSO (**c**).

**Figure 11 antioxidants-12-00273-f011:**
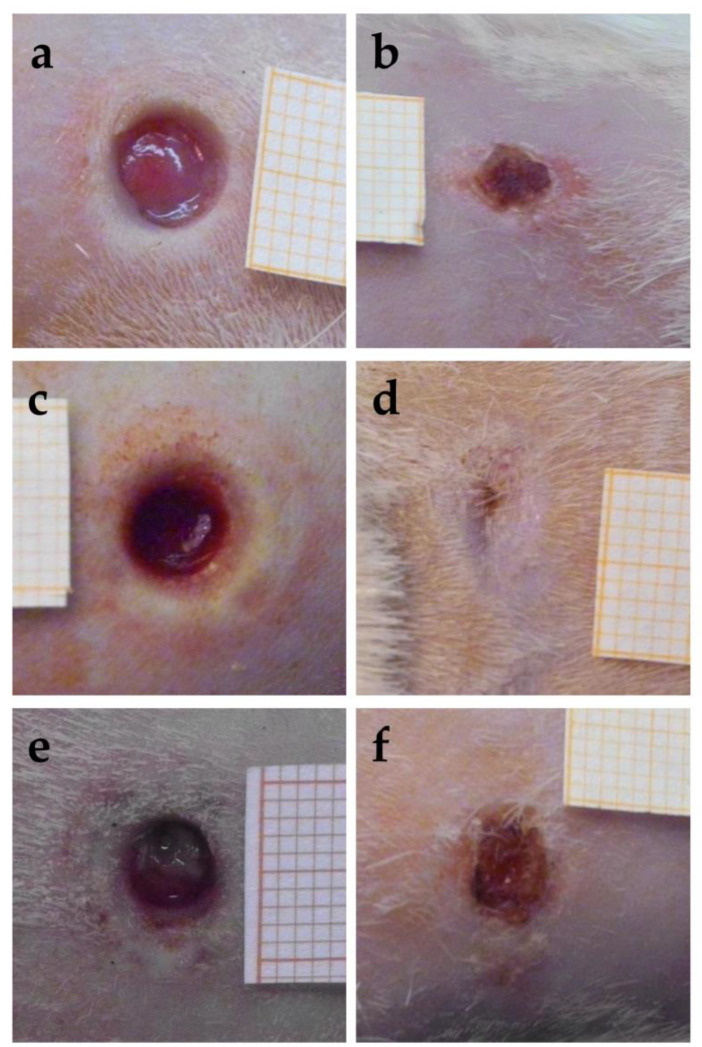
Representative images of third-grade excisional wounds 24 h after thermic injury (**a**,**c**,**e**) and after 18 days of treatment with CSO–CO 50 (**b**), CSO–CO 100 (**d**), or saline solution (NaCl, 0.9% *w*/*v*) as a reference (**f**). One square of the reference grid corresponds to 1 mm^2^.

**Figure 12 antioxidants-12-00273-f012:**
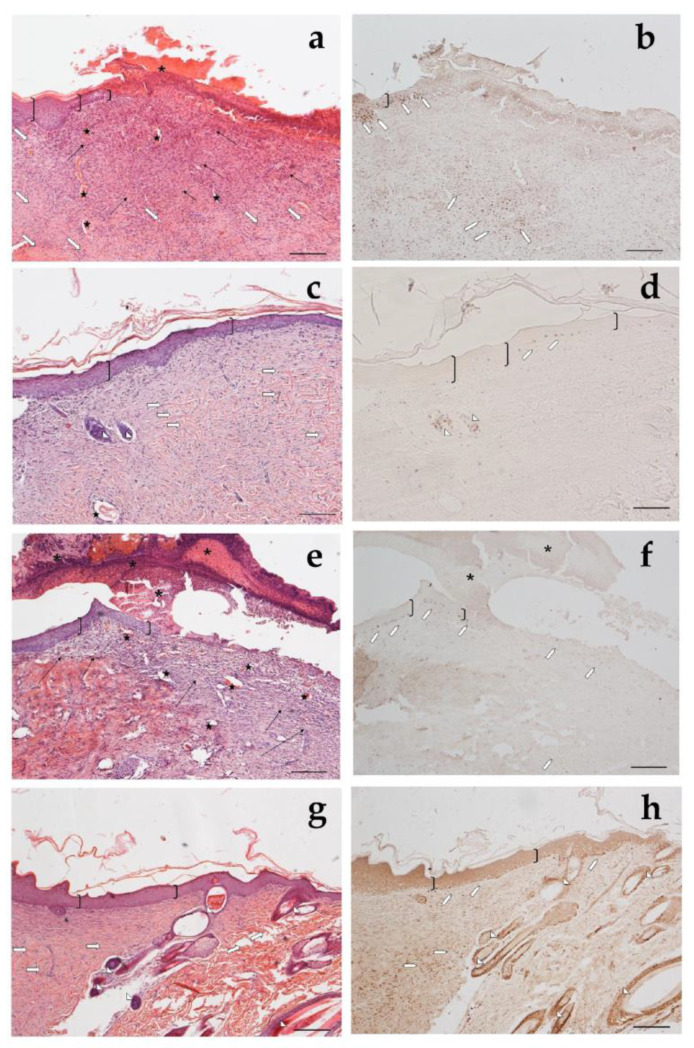
Haematoxylin and eosin staining (**a**,**c**,**e**,**g**) and BrdU sections (**b**,**d**,**f**,**h**) of skin specimens. Light microphotographs (5×) of skin sections 18 days after treatment with CSO–CO 50 (**a**,**b**), CSO–CO 100 (**c**,**d**), or saline solution (NaCl, 0.9% *w*/*v*) (**e**,**f**). Light microphotograph (5×) of intact tissue is illustrated in panel (**g**,**h**). Skin structures are labeled as follows: epidermis = bracket; vessel = star; inflammatory infiltrate = black arrows; hair follicle = triangle; organized collagen bundles = white arrow; necrotic material = asterisk; BrdU-labelled nuclei = white pentagon. Scale bar = 200 µm.

**Figure 13 antioxidants-12-00273-f013:**
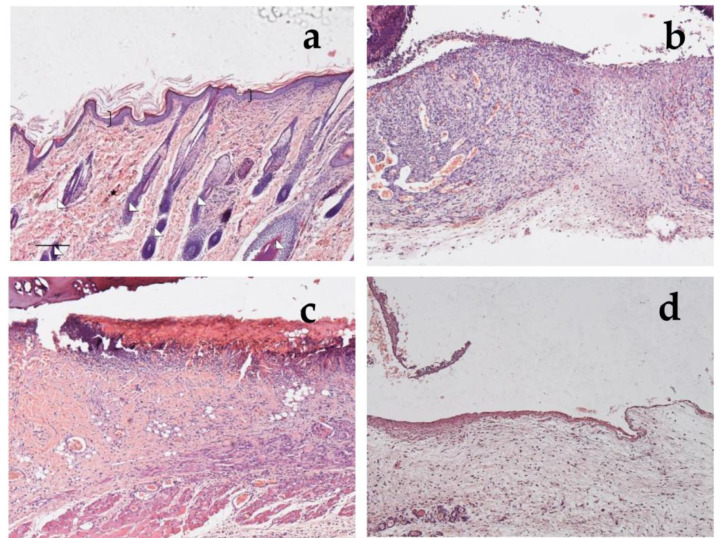
Section of healthy skin (**a**) and wounds after 7 days of treatment with saline solution (**b**), *α*-tocopherol (**c**), or CSO–CO 100 (**d**), stained with hematoxylin and eosin (bracket: pluristratified and cornified epithelium; triangles: large acidophilic bundles; star: skin appendages).

**Figure 14 antioxidants-12-00273-f014:**
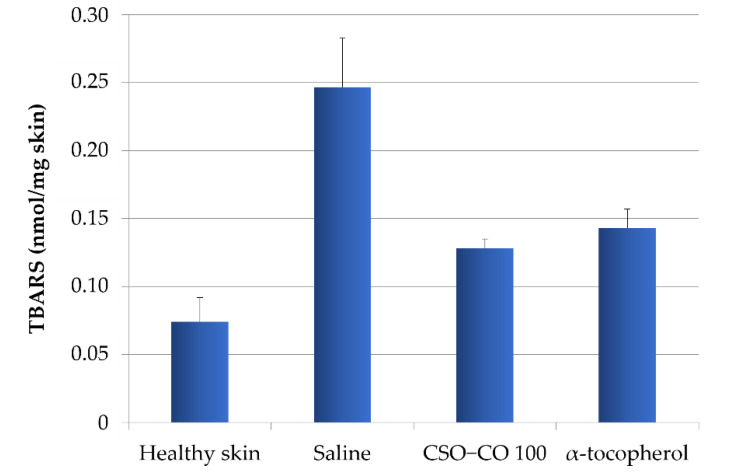
TBARS levels in healthy skin and wounds after 1 week of treatment with *α*-tocopherol and CSO–CO 100 (mean ± sd; n = 3).

**Table 1 antioxidants-12-00273-t001:** Dimensions and polydispersion index (PI) of NEs (mean ± sd; n = 3).

Formulation	Mean Number Diameter (nm)	Mean Volume Diameter (nm)	PI
CSO–CO 50	83 ± 12	220 ± 2	0.226 ± 0.048
CSO–CO 100	79 ± 38	242 ± 25	0.208 ± 0.001

## Data Availability

Data supporting the reported results are contained within the article or are available from the corresponding author.

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
