# Peer review of "Nanoemulsions of Clove Oil Stabilized with Chitosan Oleate—Antioxidant and Wound-Healing Activity"

_antioxidants, 2023, doi:10.3390/antiox12020273_

Round 1

Reviewer 1 Report

The manuscript entitled “Nanoemulsions of clove oil stabilized with chitosan oleate. Antioxidant and wound healing activity”, authored Perteghella et al., deals with the possibility of using clove oil in wound healing and assessing its antioxidant and wound healing activity. The topic of this manuscript is important and current, and results could be interesting for readers. However, some changes have to be entered into the revised version of the manuscript before it can be further processed:

1.       Line 68 – eugenol was formulated? The title refers to CO.

2.       Section 2.4 - What was the calibration curve? Was it all CO or some specific compound?

3.       In chapter 2.5 there is information about formulation with HEC. A detailed description of the preparation of the formulation should be presented in a separate chapter.

4.       Chapter 2.15 should contain a detailed description of which statistical tests were used.

5.       No study confirms the eugenol content of CO. This should be supplemented.

6.       The novelty of research should be more pointed out.

7.       The Discussion section, which is a mandatory part of the manuscript, is missing. This must be supplemented.

Author Response

Response to Reviewer 1 Comments

  1. Line 68 – eugenol was formulated? The title refers to CO.

Our work involved encapsulation of all Clove oil, whose main component is  Eugenol, and the literature cited regarding both ingredients.

  1. Section 2.4 - What was the calibration curve? Was it all CO or some specific compound?

Calibration curve was performed using the whole CO. We corrected the section 2.4

  1. In chapter 2.5 there is information about formulation with HEC. A detailed description of the preparation of the formulation should be presented in a separate chapter.

We added the requested paragraph with more details about the preparation.

  1. Chapter 2.15 should contain a detailed description of which statistical tests were used.

We added more details about the statistical tests used in the section 2.15

  1. No study confirms the eugenol content of CO. This should be supplemented.

CO was used as obtained from Sigma Aldrich. We added the lot number in the materials to allow better identification. Literature describes Eugenol as the main component of CO, and in UV spectrum of the oil solutions we obtained an absorbance maximum at 280 nm, wavelength of absorption of Eugenol (according to literature). Our aim in the analysis was to check the amount of CO in NEs at time zero and in the following weeks. We did not perform an analytical characterization of the oil, as it was beyond the aim of the work. 

  1. The novelty of research should be more pointed out.

We added some aspects that we think of novelty, in the aim part of the introduction

  1. The Discussion section, which is a mandatory part of the manuscript, is missing. This must be supplemented.

We added more details in the Results and discussion session.

Reviewer 2 Report

The paper is well structured and presented, but I have a few observations to improve the quality of the paper:

# Rewriting the summary. In its current form it looks like part of the introduction.

# Chapter 1 - Introduction - requires small improvements, by presenting other studies from the literature regarding:

- The use of essential oils (e.g. Clove oil, cannabis oil, etc.) encapsulated in various types of polymers with regenerative potential and wound healing ability (doi: 10.3390/molecules26092491 (2021), doi: 10.1038/s41598-022-23506 -0 (2022);

- The use of fatty acids for healing skin wounds and the possibility of using them as a therapeutic alternative for treating skin inflammation (doi.org/10.3390/nu14112245 (2022), doi: 10.1016/j.jep.2020.113486 (2021); Novel Clove Essential Oil Nanoemulgel Tailored by Taguchi's Model and Scaffold-Based Nanofibers: Phytopharmaceuticals with Promising Potential as Cyclooxygenase-2 Inhibitors in External Inflammation (2020) https://doi.org/10.2147/IJN.S246601.

- The authors should clearly emphasize the main novelty of this paper.

- Line 100 the abbreviation DPPH appears for the first time, and the explanation is on line 113

- The phrase on line 135-136 must be reformulated

# Chapter 2 presents in detail:

- The synthesis stages of nanoemulsions (Chitosan oleate salt (CSO), clove oil nanoemulsions (CO NE), chitosan-oleate/clove oil NE (CSO-CO NE);

- The characterization/test methods covered indicate a comprehensive summary of each technique, many details and test conditions;

- I propose to include a structural characterization of the nanoemulsions to support the chemical stability that the authors assert only through physical characterizations.

- MTT colorimetric assay should be better explained in subchapter 2.8

# Chapter 3:

- Line 527-530, Figure 10 must be replaced;

- Row 537-540, Figure 11 belongs to subchapter 3.7, and it is necessary to include it in the subchapter;

- Line 617-618, Figure 13 is not clear.

 # The conclusions are presented briefly, and the authors should detail the results that support the importance of the obtained nanoemulsions.

Author Response

# Rewriting the summary. In its current form it looks like part of the introduction.

Thank you for the suggestion, we have revised the summary.

# Chapter 1 - Introduction - requires small improvements, by presenting other studies from the literature regarding:

- The use of essential oils (e.g. Clove oil, cannabis oil, etc.) encapsulated in various types of polymers with regenerative potential and wound healing ability (doi: 10.3390/molecules26092491 (2021), doi: 10.1038/s41598-022-23506 -0 (2022);

- The use of fatty acids for healing skin wounds and the possibility of using them as a therapeutic alternative for treating skin inflammation (doi.org/10.3390/nu14112245 (2022), doi: 10.1016/j.jep.2020.113486 (2021); Novel Clove Essential Oil Nanoemulgel Tailored by Taguchi's Model and Scaffold-Based Nanofibers: Phytopharmaceuticals with Promising Potential as Cyclooxygenase-2 Inhibitors in External Inflammation (2020) https://doi.org/10.2147/IJN.S246601.

We have modified the text, adding the proposed references.

- The authors should clearly emphasize the main novelty of this paper.

We added some aspects that we think of novelty in the aim part of the introduction.

- Line 100 the abbreviation DPPH appears for the first time, and the explanation is on line 113

Thank you, we corrected in line 100.

- The phrase on line 135-136 must be reformulated

We tried to make the sentence more clear.

# Chapter 2 presents in detail:

- The synthesis stages of nanoemulsions (Chitosan oleate salt (CSO), clove oil nanoemulsions (CO NE), chitosan-oleate/clove oil NE (CSO-CO NE);

- The characterization/test methods covered indicate a comprehensive summary of each technique, many details and test conditions;

- I propose to include a structural characterization of the nanoemulsions to support the chemical stability that the authors assert only through physical characterizations.

We thank the reviewer for these suggestions.

The section of Materials and Methods was revised and implemented with a more details.

The aim of our study was to define the physical stability and the biological properties of final formulations for in vivo applications. The chemical stability of nanoemulsions was not evaluated in this study but could be considered for future studies. For this research, we evaluated the content of all clove oil after nanoemulsion preservation using spectrophotometric analysis.

- MTT colorimetric assay should be better explained in subchapter 2.8    

We tried to make the subchapter more clear.

# Chapter 3:

- Line 527-530, Figure 10 must be replaced;

The figure was replaced with a new one with high resolution.

- Row 537-540, Figure 11 belongs to subchapter 3.7, and it is necessary to include it in the subchapter;

Thank you for the suggestion, we have moved the figure

- Line 617-618, Figure 13 is not clear.

The caption of the figure was modified.

 # The conclusions are presented briefly, and the authors should detail the results that support the importance of the obtained nanoemulsions.

The text and the conclusion have been modified and implemented according to the referee suggestion.

Reviewer 3 Report

The manuscript is focused on nanoemulsions of clove oil stabilized with chitosan oleate. Clove oil is a powerful antioxidant essential oil with known anti-inflammatory, anesthetic and anti-infective properties. Such biological activities make it a promising candidate for the application in wound healing, especially in the case of chronic or diabetic wounds or burns, where the unbalance of reactive oxygen species production and detoxification is altered. 

The manuscript is interesting since it addresses the need for essential oils to tune suitable formulations to be efficiently administered in moist wounds environments. Specifically, chitosan hydrophobically modified by ionic interaction with oleic acid was used in the present work to stabilize clove nanoemulsions. Antioxidant properties of were evaluated in vitro by DPPH assay, and in fibroblast cell lines by Electron Paramagnetic Resonance using α-phenyl-N-tert-butyl nitrone as spin trap. Also,a burn murine model was used to evaluate the macroscopic wound closure and the skin reparation. 

The paper is overall well-written. My major concern is related to the fact that biological activity results must be supported by chemical characterization of the plant material used in the study. In this context, the authors may employ GC coupled to FID or MS detection. If FID will be used, simple GC percentages obtained can be used assuming response factors equal to unity for all the components. If MS detection will be performed,  this can achieved by means of reference materials. 

Author Response

The manuscript is focused on nanoemulsions of clove oil stabilized with chitosan oleate. Clove oil is a powerful antioxidant essential oil with known anti-inflammatory, anesthetic and anti-infectiveproperties. Such biological activities make it a promising candidate for the application in wound healing, especially in the case of chronic or diabetic wounds or burns, where the unbalance of reactive oxygen species production and detoxification is altered.

The manuscript is interesting since it addresses the need for essential oils to tune suitable formulations to be efficiently administered in moist wounds environments. Specifically, chitosan hydrophobically modified by ionic interaction with oleic acid was used in the present work to stabilize clove nanoemulsions. Antioxidant properties of were evaluated in vitro by DPPH assay, and in fibroblast cell lines by Electron Paramagnetic Resonance using α-phenyl-N-tert-butyl nitrone as spin trap. Also, a burn murine model was used to evaluate the macroscopic wound closure and the skin reparation.

The paper is overall well-written.

My major concern is related to the fact that biological activity results must be supported by chemical characterization of the plant material used in the study. In this context, the authors may employ GC coupled to FID or MS detection. If FID will be used, simple GC percentages obtained can be used assuming response factors equal to unity for all the components. If MS detection will be performed, this can achieved by means of reference materials.

Thank you for the suggestion. Our work was aimed especially at overcoming the challenges that EO formulation presents, due to their sensitivity to environmental conditions and especially to their volatility. NE were selected to maximize the interaction of the oil with the wound site. We assumed the total oil as reference to study the formulation behaviour, although we are aware that the composition is very complex. Clove oil composition, obtained by GC-MS has been recently published (Hameed et al, Molecules, 2021), on a clove oil sample coming from the same supplier of our material. According to other studies, they found in it mainly eugenol and lower amount of caryophyllene, for a total of 33 components. Considering the complexity of the analysis, that requires the availability of the reference standards, we did not repeat it

In the revised version, for better identification, we added the lot number of clove oil in the materials. We also referred more clearly to the literature for more extensive chemical characterization.

Reviewer 4 Report

The article "Nanoemulsions of clove oil stabilized with chitosan oleate. Antioxidant and wound healing activity" describes the wound dressing capabilities of a new formulation based on chitosan and essential oil. It is a valuable study that can be published after authors address the following problems:

Definition for nano-systems is to have at least one dimension under 100 nm. In this case the diameter of the particles is larger than 100 nm therefore the term nanoemulsion does not reflect the truth.

Plant names should be written with italics (e.g. row 36).

I would recommend use the term loading efficiency instead of loading yield.

Following literature could prove the synergic activity of chitosan and essential oils as described in this manuscript doi: 10.3390/foods9121801; doi: 10.3390/pharmaceutics13111939; doi: 10.3390/pharmaceutics13020195

Top right of the all the pages in manuscript have 5/5.

Please improve the quality of some figures: e.g. fig. 14; fig. 10 is cut on the 10c/right and could have axes and number clearer.

How is this system a better one? Conclusion section must be reworked to underline the novelty and advantages of this research, with actual numbers. The conclusion should reflect the heuristic of the study.

Author Response

Definition for nano-systems is to have at least one dimension under 100 nm. In this case the diameter of the particles is larger than 100 nm therefore the term nanoemulsion does not reflect the truth.

From the EU definition of nanomaterial, you are perfectly right that dimensions should be lower than 100 nm: “Nanomaterial” means a natural, incidental or manufactured material containing particles, in an unbound state or as an aggregate or as an agglomerate and where, for 50% or more of the particles in the number size distribution, one or more external dimensions is in the size range 1 nm - 100 nm”. The dimensional distribution we assessed was however a volume distribution, that is more sensitive to the presence of larger particles and allows to quickly understand the increase of dimensions for aggregation, or in the case of dispersions, coalescence. The number distribution gives a mean value much lower.

Plant names should be written with italics (e.g. row 36).

Thank you, we corrected line 36.

I would recommend use the term loading efficiency instead of loading yield.

We have corrected the text and the figure 3 according to the suggestion

Following literature could prove the synergic activity of chitosan and essential oils as described in this manuscript doi: 10.3390/foods9121801; doi: 10.3390/pharmaceutics13111939; doi: 10.3390/pharmaceutics13020195

Top right of the all the pages in manuscript have 5/5.

This should occur during the uploading of the manuscript.

Please improve the quality of some figures: e.g. fig. 14; fig. 10 is cut on the 10c/right and could have axes and number clearer.

The figures have been modified.

How is this system a better one? Conclusion section must be reworked to underline the novelty and advantages of this research, with actual numbers. The conclusion should reflect the heuristic of the study.

The text and the conclusion have been modified to underline the novelty and the advantages of our proposed formulation.

Reviewer 5 Report

The methodological design of the research is good. The paper is well written. 

A more detailed description of statistical methods is recommended.

Author Response

Response to Reviewer 5 Comments

Point 1: The methodological design of the research is good. The paper is well written.

A more detailed description of statistical methods is recommended.

Response 1: We added more details about the statistical tests used in the section 2.15

Round 2

Reviewer 1 Report

I'm still not convinced by the eugenol content of CO. Without experimental data, it is impossible to assess and claim that CO has the highest eugenol content. UV is not a reference method, especially since it is known that there are more compounds in CO than just eugenol. UV gives a signal from all compounds and is not a selective method. I believe that without reliable analytical research, the work does not meet the requirements.

Author Response

Response to Reviewer 1 Comments

I'm still not convinced by the eugenol content of CO. Without experimental data, it is impossible to assess and claim that CO has the highest eugenol content. UV is not a reference method, especially since it is known that there are more compounds in CO than just eugenol. UV gives a signal from all compounds and is not a selective method. I believe that without reliable analytical research, the work does not meet the requirements.

We understand your concern, but we need to point out that:

1) For Eugenol as main component in Clove Oil we have referred to the literature [ Chaieb et al,  Phytother. Res. 21, 501–506 (2007) and  Jirovetz et al, J. Agric. Food Chem. 2006, 54, 6303-6307], where the eugenol content was found 88 and 77 % respectively.  In particular we referred to the reference [1 Hameed, M.; Rasul, A.; Waqas, M.K.; Saadullah, M.; Aslam, N.; Abbas, G.; Latif, S.; Afzal, H.; Inam, S.; Shah, P.A. Formulation and Evaluation of a Clove Oil-Encapsulated Nanofiber Formulation for Effective Wound-Healing. Molecules 2021, 26, doi:10.3390/molecules26092491]; in which the GC-MS analysis was recently carried out on a clove oil sample obtained from the same supplier of the CO we used. According to the previous literature, these authors found the composition of the table below (Eugenol 71.4%; Eugenyl Acetate 8.32%). The authors of this article used and analysed the same commercial Clove Oil that we purchased for our research study, so we considered the Eugenol as the main component of Clove Oil.

On this basis, in our literature survey and in our discussion we considered not only articles where CO is studied but also articles in which eugenol is evaluated.

2) It must be considered that the main interest in our work was not to attribute the effects we observed to Eugenol, but to develop a new formulation for Clove oil, and to confirm that in our formulation Clove oil maintained the antioxidant activity, anti-inflammatory and wound healing effects that are described in the literature (and in the literature mainly attributed to eugenol). To assess the physical stability, we considered the initial CO amount as 100% and we always referred to that as reference. The evaluation of chemical stability was not in the aim of our study.

However, we found in the literature other authors that used the UV method to quantify the Clove Oil, for example the article titled “Complexation of Eugenol (EG), as Main Component of Clove Oil and as Pure Compound, with β- and HP-β-CDs” (Food and Nutrition Sciences, 2012, 3, 716-723; DOI: 10.4236/fns.2012.36097) compared Capillary GC-MS Analysis and UV-Spectrophotometric Analyses, where the reliability of this last assay was claimed.

Reviewer 2 Report

In my opinion, the article can be published in its current form.

Author Response

Thanks for your positive feedback.

Reviewer 3 Report

The authors have adequately addressed all remarks and the paper can be now accepted in the present form. 

Author Response

Thanks for your positive feedback.

Round 3

Reviewer 1 Report

 Accept in present form